# Interplay between BMPs and Reactive Oxygen Species in Cell Signaling and Pathology

**DOI:** 10.3390/biom9100534

**Published:** 2019-09-26

**Authors:** Cristina Sánchez-de-Diego, José Antonio Valer, Carolina Pimenta-Lopes, José Luis Rosa, Francesc Ventura

**Affiliations:** 1Departament de Ciències Fisiològiques, Universitat de Barcelona, Carrer Feixa Llarga s/n, 08907 L’Hospitalet Llobregat, Spain; csanchezdg@gmail.com (C.S.-d.-D.); j.a.valer@hotmail.com (J.A.V.); carolinapimentacosta@ub.edu (C.P.-L.); joseluisrosa@ub.edu (J.L.R.); 2IDIBELL, Avinguda Granvia de l’Hospitalet 199, 08908 L’Hospitalet de Llobregat, Spain

**Keywords:** BMP, reactive oxygen species (ROS), NOX, cell signaling, PI3K, MAPK, SMAD, NRF2

## Abstract

The integration of cell extrinsic and intrinsic signals is required to maintain appropriate cell physiology and homeostasis. Bone morphogenetic proteins (BMPs) are cytokines that belong to the transforming growth factor-β (TGF-β) superfamily, which play a key role in embryogenesis, organogenesis and regulation of whole-body homeostasis. BMPs interact with membrane receptors that transduce information to the nucleus through SMAD-dependent and independent pathways, including PI3K-AKT and MAPKs. Reactive oxygen species (ROS) are intracellular molecules derived from the partial reduction of oxygen. ROS are highly reactive and govern cellular processes by their capacity to regulate signaling pathways (e.g., NF-κB, MAPKs, KEAP1-NRF2 and PI3K-AKT). Emerging evidence indicates that BMPs and ROS interplay in a number of ways. BMPs stimulate ROS production by inducing NOX expression, while ROS regulate the expression of several BMPs. Moreover, BMPs and ROS influence common signaling pathways, including PI3K/AKT and MAPK. Additionally, dysregulation of BMPs and ROS occurs in several pathologies, including vascular and musculoskeletal diseases, obesity, diabetes and kidney injury. Here, we review the current knowledge on the integration between BMP and ROS signals and its potential applications in the development of new therapeutic strategies.

## 1. Introduction

Bone morphogenetic proteins (BMPs) were first described in the 1960s as osteoinductive soluble factors that belong to the transforming growth factor-β (TGF-β) superfamily [1]. BMPs are expressed in multiple cell types and participate in a wide variety of processes, including morphogenesis, cell differentiation and regulation of whole-body homeostasis. Several studies demonstrate that BMPs contribute to gastrulation and mesoderm patterning. In extraembryonic tissues, BMPs are critical for primitive streak formation [2] and to establish left-right pattering [3]. During organogenesis, BMP signaling promotes mesoderm differentiation into heart tissue. In the ectoderm, BMPs induce neural differentiation and are necessary for neural tube formation [4]. In the peripheral nervous system, BMPs induce neuronal lineage commitment and neuronal differentiation whereas in the progenitor cells of the Central Nervous System (CNS), BMPs suppress neuronal or oligodendroglial differentiation and promote astroglial formation [5]. BMPs are also involved in skeletogenesis [6] and bone remodeling after birth [7]. In fact, BMP signaling is required for chondrogenesis [8] and osteogenesis [9]. In kidney organogenesis, BMPs promote the elongation of the ureteric bud and regulate the proliferation and differentiation of the metanephric mesenchyme [10]. BMP signaling is also critical for the development of vascular system in embryos, and for the regulation of vascular homeostasis in adults and, when altered, being causative for several vascular dysfunctions. For instance, pulmonary arterial hypertension and hereditary hemorrhagic telangiectasia have been directly linked to impaired BMP signaling [11,12,13]. Finally, BMPs are crucial for a proper reproductive system function. In gonadal differentiation, BMP8 is important for spermatogenesis maintenance [14], while BMP15 is associated with granulosa cell proliferation in the ovary [15]. Due to BMP relevance in organ generation and maintenance, alterations in BMP signaling are associated with several pathologies.

BMPs interact with specific serine-threonine kinase receptors present in the cell membrane. Subsequently, the information is transduced to the nucleus through the SMAD pathway. Besides this signaling mode, several non-canonical, SMAD-independent BMP transducers have been identified, including small GTPases, phosphatidylinositol 3-kinase/AKT (PI3K/AKT) and distinct types of mitogen-activated protein kinases (MAPKs) [16]. In this context, recent evidence suggests that reactive oxygen species (ROS) can also act as second messengers for BMPs.

ROS are highly reactive molecules derived from the reduction of molecular oxygen as a consequence of cellular metabolism or the activity of specific enzymes, such as the NADPH oxidase (NOX) complexes [17]. Since ROS are highly unstable, they react with proteins, nucleic acids, lipids and other cellular components and often disrupt their cellular functions. Therefore, ROS were first considered as harmful molecules. Accordingly, oxidative stress is involved in multiple pathologies such as cardiac and neurodegenerative diseases, vascular disorders, diabetes, infections and cancer [18]. Even though oxidative stress occurs in several pathologies, numerous studies demonstrate that ROS signaling is also important in normal physiology and in the generation of proper redox biological responses [19]. For instance, in skeletal muscle, ROS are required for normal contraction, where they modulate muscle adaptations to exercise [20]. ROS also contribute to immune response activation and regulation [21] and mediate leucocyte adhesion to endothelial cells [22]. Indeed, the physiological role of ROS is based on their capacity to modulate several signaling pathways, including nuclear factor-kappa B (NF-κB), MAPKs, kelch-like ECH-associated protein 1 (KEAP1)-nuclear factor erythroid 2-related factor 2 (NRF2) and PI3K-AKT [19].

Although little is known about the relationship between BMPs and ROS, both signals share common transducing modules, and their dysregulation occur in several pathologies. In this review, we will describe the putative modes of signaling crosstalk between BMPs and ROS and analyze their implications in physiology and disease.

## 2. BMP Signals

BMPs structurally belong to the transforming growth factor-β (TGF-β) superfamily [23]. BMPs share seven highly conserved cysteines with other TGF-β family members. Six of them form three intramolecular disulfide bonds, and the seventh one creates a covalent disulfide bond with another monomer to form homo- or heterodimers [24]. BMPs are also sub-classified into several groups, including BMP2/4, BMP5/6/7/8, growth and differentiation factor (GDF)-5/6/7 and BMP9/10 [23]. BMPs diffuse away from the cell of origin or form stable complexes with their cleaved proto-domains.

Outside the cell, BMPs bind to BMP receptors (BMPRs) and initiate an intracellular transduction cascade. BMPRs are serine/threonine kinase receptors with a short extracellular domain, a single transmembrane domain and an intracellular serine/threonine kinase domain. They are classified in two groups: BMPR type I (ACVR1, ACVRL1, BMPR1A and BMPR1B) and BMPR type II (BMPR2, ACVR2 and ACVR2B) [25]. Different BMP subgroups display disparate type I and type II receptor binding specificities [23]. BMP binding induces the formation of a heteromeric complex between type I and type II receptors. Once the complex is established, constitutively active kinase activity present in type II receptors phosphorylates the GS domain (a glycine and serine-rich intracellular region located adjacent to the kinase domain) of type I receptors, an action that leads to its activation. The active complex phosphorylates and activates the SMAD family of transcription factors that transduce the signal to the nucleus [26]. Under basal conditions, receptor-regulated SMADs (R-SMAD) are anchored at the cell membrane through their interaction with several proteins. In the canonical pathway, type I receptors phosphorylate R-SMADs 1/5/8. R-SMAD phosphorylation releases them from the membrane and disrupts the auto-inhibitory interaction between their N- and C-terminal domains. Then, two active R-SMAD 1/5/8 molecules bind to the common partner SMAD4 and translocate to the nucleus, where they interact with several transcription factors and transcriptional co- activators/repressors [27,28] (Figure 1).

Besides the canonical signaling through SMADs, BMPs modulate other intracellular signaling pathways with important roles in cell physiology. Non-canonical BMP signaling includes small GTPases, PI3K/AKT, lim kinase-1 (LIMK1) and various types of MAPKs [28]. For instance, BMPs regulate p38 activity through TGF-β-associated kinase 1 (TAK1) [29,30]. TAK1 is a member of the MAPKKK family whose activity is regulated by the E3 ubiquitin ligase TRAF6 in a receptor-kinase-independent manner. Upon BMP/BMPR complex formation, TRAF6 becomes poly-ubiquitinated and recruits and triggers TAK1 activation that induces p38 activity [31] (Figure 1).

BMP signaling is regulated by multiple mechanisms. Intracellularly, several miRNAs regulate the expression of BMPs, BMPRs and downstream effectors. Inhibitory Smads (such as Smad 6 and 7) and methylation of the promoters also regulate BMP signaling. Extracellularly, pseudo-receptors and BMPs antagonists control BMP availability [32,33]. More than 15 antagonists have been discovered which are classified in three subgroups based on the size of the cysteine knot, a common feature of BMPs. First, antagonists with an eight-member ring represented by the DAN family (including DAN, GREM2, Gremlin, Cerberus, Coco and Caronte. second, USAG-1, DANTE and twisted gastrulation (TSG) with a nine-member ring and finally, inhibitors with a ten-member rings including Chordin (CHD), Noggin (NOG), Ventroptin (CHRDL1), Follistatin (FST) and FLRG-follistatin-related gene (FSTL3) [32,34]. Finally, the binding of BMPs to several extracellular matrix (ECM) proteins also modulates their bioavailability [34].

## 3. ROS

ROS are radical and non-radical oxygen species that result from the partial reduction of oxygen [35]. Cellular ROS include the superoxide anion (O_2_^−^), hydrogen peroxide (H_2_O_2_), and hydroxyl radical (OH·) that arise either as byproducts of aerobic metabolism or as defense mechanisms against xenobiotics or bacterial invasion [36]. Under physiological conditions, ROS act as signaling molecules that are important for adaptation to changes in nutrient supply and the oxidative environment [37]. ROS also contribute to regulate cell proliferation and differentiation. However, increased ROS levels or changes in their compartmentalization can dysregulate oxidation of proteins, lipids and nucleic acids [37]. Therefore, the ROS concentration is in a dynamic equilibrium and is modulated by cellular processes that produce and eliminate ROS.

A primary ROS source is O_2_^−^, which is generated in the mitochondria by complexes I, and III or in other cellular compartments by the action of several enzymes such as NOXs. Superoxide reacts with molecules to cause molecular damage or turn into H_2_O_2_ spontaneously or by the action of the enzyme superoxide dismutase (SOD). Other ROS sources are the mitochondrial enzymes monoamine oxidase and cytochrome b5ductase, as well as glycerol-3-phosphate dehydrogenase, aconitase, pyruvate dehydrogenase or α-ketoglutarate dehydrogenase [38].

Since high ROS levels generate cell damage, cells have developed several defense mechanisms to detoxify ROS. H_2_O_2_ is reduced to generate water by the action of catalase or glutathione peroxidase (GPx). Thioredoxins, peroxiredoxins and GPxs act as antioxidant enzymes by reversing the oxidation of cysteine residues. Finally, cells also have non-enzymatic antioxidants that include low-molecular-weight compounds, such as vitamins C and E, β-carotene, uric acid and glutathione (GSH) [36].

## 4. ROS-Mediated Cellular Signaling

ROS are highly reactive molecules with the capacity to target several transductor proteins (NF-κB, MAPKs, KEAP1-NRF2 and PI3K-AKT), ion channels and transporters (Ca^2+^ and mitochondrial permeability transition pore [mPTP]), and thus, impact on cell growth, differentiation, migration and death [37] (Figure 2).

The NF-κB family of transcription factors are involved in multiple cellular processes such as immune and inflammatory responses, development, cellular growth and apoptosis. NF-κB dimers are bound to their inhibitors of κB (IκBs) and retained in the cytoplasm. Consequently, NF-κB activation depends on IκB degradation, which occurs after its phosphorylation by IκB kinase (IKK). Activators of the IKK complex include MAPKKKs such as MEKK1, MEKK3 and TAK1, which represent a common hub for several stimuli, including BMPs and ROS. Recent studies demonstrate that ROS can induce IκBα phosphorylation and degradation [39]. ROS also induce S-glutathionylation and IKK inhibition [40] and other upstream kinases, such as MKK1; these actions hamper NF-κB activation. Conversely, NF-κB can alter ROS levels by increasing the expression of antioxidant proteins such as Cu-Zn-SOD, Mn-SOD, GPx and glutathione S-transferase-Pi (GST-Pi) [37].

ROS can stimulate MAPK pathways at different levels. ROS activate growth factor receptors even without binding to their corresponding ligand [41]. ROS also mediate the oxidation of Cys124 in extracellular signal-regulated kinase (ERK), an action that results in its activation [42]. ROS can also activate the ERK pathway indirectly when superoxide induces phospholipase C-γ (PLC-γ) phosphorylation, which generates inositol trisphosphate (IP3) and diacylglycerol (DAG) [43]. c-Jun N-terminal kinase (JNK) is another MAPK activated by cytokines (tumor necrosis factor [TNF] and FAS) and several environmental stresses such as oxidative stress. In this case, MAPKKs phosphorylate JNK on critical threonine and tyrosine residues, changes that lead to its activation. Under unstressed conditions, apoptosis signal-regulating kinase 1 (ASK1), a MAPKKK in the pathway, is bound to thioredoxin (TRX), which inhibits its activity. TRX oxidation by ROS induces ASK1 release and activation [44]. The p38 MAPK pathway is also stimulated by cellular stress and cytokines, such as TNF-α and interleukin-1beta (IL-1β). ROS can induce p38 pathway in several ways. As mentioned previously, ROS can directly activate growth factor receptors. Oxidative stress also directly or indirectly affect MAPKKKs involved in p38 signaling, including MEKKs and MLK3 [37].

PI3K-AKT is also susceptible to ROS regulation. PI3K catalyzes phosphatidylinositol 3,4,5-trisphosphate (PIP3) synthesis that recruits it into the plasma membrane and activates proteins that contain the pleckstrin homology domains such as AKT and its upstream activator PDK1. Activation of AKT is induced by its phosphorylation in Thr308 and Ser473 [45]. AKT activity is counteracted by its dephosphorylation by protein phosphatase 2 (PP2A). PP2A is involved in the regulation of WNT, mTOR and MAP kinase pathways, cell cycle progression as well as DNA repair [46]. Inhibition of mTORC1 activates PP2A and subsequent DNA-PK. DNA-PK is required for the repair of DNA double-strand breaks (DSBs). Besides its role in DNA repair, DNA-PK also phosphorylates AKT in Ser473 kinase, under DNA damage conditions [47]. On the other hand, AKT activity can be also reduced through PIP3 dephosphorylation by phosphatase and tensin homolog (PTEN). ROS induce PTEN degradation [48] and may also promote oxidative inactivation of PP2A [49].

ROS also regulate protein kinase pathways through the oxidation of cysteine sulfhydryl groups that are present in several protein kinases, including protein kinase A (PKA) [50], protein kinase C (PKC) [51], protein kinase D (PKD) [52], receptor tyrosine kinases [53] and Ca/calmodulin-dependent protein kinase II (CaMKII) [54]. Recently, a study demonstrated that ROS oxidize Cys17 and 38 of the PKA regulatory subunits, changes that lead to the formation of an inter-subunit disulfide bond that activates PKA independently of cyclic AMP (cAMP) levels [55]. The PKC family contains membrane protein kinases composed of an N-terminal regulatory region and a C-terminal catalytic domain [56]. Both domains contain cysteine-rich motifs targeted by ROS and they play a dual role in PKC regulation. In the regulatory domain, these cysteines form two pairs of zinc fingers that inhibit PKC activity. Low ROS doses destroy the zinc finger conformation, thus activating PKC even in the absence of Ca^2+^ or phospholipids. However, high ROS doses react with cysteine residues in the catalytic domain, and these interactions lead to PKC inactivation [57,58]. Additionally, ROS enhance PKD activation in several ways: they induce the production of DAG, the activation of PKC and the phosphorylation of PKD at Tyr93, which facilitates its binding with PKC [59,60].

The KEAP1-NRF2 pathway is essential for the maintenance of redox balance. Under basal conditions, NRF2 remains in the cytosol associated with KEAP1, which recruits CUL3 that ubiquitinates NRF2, leading to its degradation. Increased ROS levels promote the dissociation of the NRF2-KEAP1 complex by the oxidation of key reactive cysteine residues (Cys151, Cys273 and Cys288) of KEAP1 and/or the activation of kinases such as PKC, PI3K or AMPK that phosphorylate NRF2 [61,62,63,64]. Both events reduce NRF2 degradation and facilitate its translocation to the nucleus. Once in the nucleus, NRF2 heterodimerizes with other transcription factors and binds to antioxidant response element (ARE) sequences to promote the expression of target genes [65]. Activation of these different pathways also contributes to ROS scavenging through the induction of NRF2. NF-κB acts as a transcription factor for NRF2 [66] and, as mentioned above, PKC and PI3K phosphorylate NRF2, an action that stabilizes the protein and promotes its nuclear translocation [62,63,64]. Simultaneously, ROS activate glycogen synthase kinase 3β (GSK3β), which leads to nuclear import of Src kinases that phosphorylate NRF2 at Tyr568 and promote its nuclear export [67].

## 5. NOXs

NOXs and dual oxidases (DUOXs) are membrane-bound enzymatic complexes that catalyze the transfer of one electron from NADPH to oxygen to generate ROS in response to a stimulus, such as growth factors, cytokines and calcium signals. In the human genome, seven *NOX* homologues have been identified: *NOX1* to *NOX5* and *DUOX1* and *DUOX2*. These genes differ in their preferential tissue of expression, the type of ROS released and their regulatory mechanisms. NOXs produce either superoxide (NOX1-3 and NOX5) or H_2_O_2_ (NOX4 and DUOX1-2) in a NADPH-dependent manner [68]. NOXs are classified in three main groups according to the domains they contain. NOX1, NOX2, NOX3 and NOX4 contain the electron transfer center. NOX5 contains the catalytic domain and an amino-terminal calmodulin-like domain that acts as a regulatory domain. DUOXs contain the catalytic domain and a calmodulin-like domain connected by an α-helix to an amino-terminal peroxidase-homology domain.

Short-term NOX regulation involves regulatory subunits. NOX1, NOX2 and NOX3 are present as inactive transmembrane monomers, and interaction with other subunits is required for their activity. However, NOX4 is synthesized in a constitutively active form [69]. NOX1 to NOX4 interact with p22phox, which has a scaffold function. NOX5 does not interact with p22phox but requires homo- or multimerization for its stabilization [70] and depends on cytosolic calcium levels for its activation. To activate the NOX catalytic center, three signaling events must occur. First, the regulatory domain (p40phox, p47phox and p67phox), which is in the cytosol, is phosphorylated in the auto-inhibitory region. This process involves protein kinases, including PKC and AKT [71]. 

Then, phosphorylation of p47phox releases its binding to the SRC-homology 3 (SH3) domain, allowing p47phox to bind to p22phox (in the cell membrane) and to lipids provided by PI3K and phospholipase D [72,73]. Finally, it is required the presence of two guanine nucleotide-binding proteins, RAP1A and RAC2. RAP1A is a membrane protein, while RAC2 localizes in the cytosol in a dimeric complex with its inhibitor guanine nucleotide dissociation inhibitor (Rho-GDI). Upon its activation, RAC2 binds GTP and translocates to the membrane along with p40phox, p47phox and p67phox, that constitute the active complex [74].

NOX enzymes are also regulated at the transcriptional level. This regulation depends on the specific enzyme and cell type. For instance, *NOX1* is strongly induced by the growth factors angiotensin II, platelet-derived growth factor (PDGF) and phorbol esters (in vascular smooth muscle) and keratinocyte growth factor-alpha (KGF-α) and interferon gamma (IFN-γ) in human colon cells. Comparatively, *NOX4* is regulated by angiotensin II [75].

NOX enzymes participate in several biological processes. In neutrophils, NOXs generate high superoxide concentrations in the phagosome to enhance killing bacteria. In barrier cells, IFN-γ and lipopolysaccharide (LPS) induce *NOX1* expression through Toll-like receptor-4 to help kill invading microorganisms. In addition to host defense, NOX-derived ROS contribute to signal transduction by regulating biological processes such as cell growth, differentiation, angiogenesis and senescence. For instance, *NOX4* overexpression in tumors is associated with apoptosis and cell senescence [75]. NOX4 is also implicated in angiotensin-II-dependent signaling in vascular smooth-muscle cells [76] and in insulin signaling in adipose tissues [77]. DUOX enzymes are implicated in biosynthetic reactions that involve ECM proteins [75].

## 6. Interplay between BMPs and ROS Signaling

BMPs stimulate ROS production in several cell types. BMPs stimulate ROS production by activation and/or induction of *NOX1-5* expression. For instance, in adult renal progenitors BMP2 induces *NOX4* and *NOX2* expression and activity [78]. ROS reciprocally regulate the expression of *BMP2* in osteoblasts [79]. Both BMPs and ROS activate several common signaling pathways that should be coordinated (Figure 3). Consequently, unregulated changes in their expression or production or altered signaling could result in severe repercussions.

NOX1, NOX2 and NOX3 require p47phox and p67phox phosphorylation to facilitate translocation of the cytosolic subunits and formation of an active complex. The primary kinase involved in p47 phosphorylation is PKC (α, β, δ, and ζ). Other pathways, including ERK and p38, PKA and AKT, also contribute to NOX activation [80]. Therefore, BMPs can activate NOX enzymes by inducing their expression or through phosphorylation by the non-canonical signaling pathways above mentioned. For instance, in osteoblasts BMP2 enhances ROS production by activating NOX [79], and BMP2, BMP4 and BMP7 upregulate *NOX4*, *NOX1* and *NOX2* expression in osteoblasts, sympathetic neurons and monocytes [79,81,82]. Moreover, in podocytes, BMP2 increases the free cytosolic Ca^2+^concentration, and the expression of DNA-binding protein inhibitor 1 (ID1). Both events elevate NADPH-dependent ROS production [83].

Activation of non-canonical pathways by BMPs also induces cyclooxygenase 2 (*COX2*) gene expression [84], which is a source of ROS [85]. Moreover, different prostanoids produced by COX2 modulate NOX activity and regulate ROS production [85]. Finally, PI3K/AKT signaling, which can be activated by BMPs [30], contribute to elevate ROS levels by activating NOX, modulating mitochondrial bioenergetics and enhancing the metabolic rate. In this latter case, AKT phosphorylate GSK3β, which reduces its activity thus enhancing the activity of pyruvate dehydrogenase and α-ketoglutarate dehydrogenase complexes. Both of these complexes generate superoxide and H_2_O_2_ [86]. Unlike other NOX isoforms, NOX4 is constitutively active. Thus, its regulation occurs mainly at the transcriptional level. Several cytokines, including BMP2, TGFβ and angiotensin II, can upregulate *NOX4* expression through PKC and/or SMAD pathways [78,87].

Reciprocally, ROS contribute to the regulation of BMP expression. In particular, *BMP2* and *BMP4* are upregulated in conditions associated with oxidative stress [11]. For example, *BMP2* autoregulation of its own expression is mediated by SMAD-dependent transcription and can also be modulated by NOX4-derived ROS in osteoblasts [79]. Further, increased H_2_O_2_ levels, produced by NOX activation or other conditions such as inflammation, induce NF-κΒ activation. Once active, NF-κΒ enhances *BMP2* expression [11,88]. Therefore, although the mechanisms are mostly unknown, evidence suggests that there is a positive regulatory loop between ROS and BMP signals. Consequently, alteration of these regulatory loops can lead to oxidative stress and cell dysfunction and contribute to several pathologies.

The crosstalk between BMPs and ROS goes beyond the regulation of their own transcription and activation; it also affects downstream signaling pathways shared by these two molecules. As mentioned above, ROS activate ERK, JNK and p38 MAPKs [42], while BMPs activates TAK1 that acts upstream of JNK and p38 [89]. PI3K-AKT is another intracellular pathway where ROS and BMP signaling converge. BMP2 increases AKT phosphorylation in pancreatic cancer cells [90], while ROS reduces PI3K-AKT downregulation by targeting PTEN and PP2A in embryonic rat heart cells [91].

## 7. BMPs and ROS in Cell Specification

BMPs were identified as molecules capable of modulating mesenchymal cell specification and inducing bone development. In particular, BMPs regulate osteoblast differentiation through induction of bone-determining transcription factors [92]. ROS generation also influences bone cell differentiation. Undifferentiated mesenchymal stem cells (MSCs) display low levels of ROS and express high levels of antioxidant enzymes. This evidence is linked to MSC self-renewal, as physiological ROS upregulation is required for MSC differentiation. For instance, during MSC osteogenic differentiation, WNT/β-catenin activation increases ROS [93]. In osteoblast progenitor cells, BMP2 activates *NOX4*, which is highly expressed in pre-osteoblasts, and this activation results in ROS production. Superoxide anions generated by NOX4 are rapidly converted into H_2_O_2_, and ROS produced by NOX4 induce alkaline phosphatase expression and activate PI3K and SMAD 1/5 signal transduction pathways. MSCs also generate both permanent and transient cartilage. BMP signaling is critical for cartilage formation [94], and ROS generated by NOX2 and NOX4 are also essential for the early chondrogenesis stages. ROS also contribute to osteoclast differentiation. Osteoclasts arise from the fusion of multiple monocyte and macrophage precursors in the presence of receptor activator of NF-κB ligand (RANKL). RANKL stimulates ROS production, which is required for RANKL-induced activation of AKT, NF-κB and ERK pathways [95]. Moreover, ROS are necessary for the resorptive function of osteoclasts [96].

ROS and BMPs also participate in the cardiovascular system development. In embryonic stem cells, ROS promote differentiation toward both cardiomyogenic and vascular cell lineages through AMP-activated protein kinase (AMPK) pathway activation. ROS and NOX4 activation are also involved in cardiogenesis and cardiac myofibrillogenesis, since antioxidants and NOX4 silencing impair the cardiac differentiation of stem cell precursors. In early embryonic mouse heart, *NOX4* is the main NOX isoform expressed. NOX4 generates ROS, an action that leads to p38 phosphorylation and monocyte enhancer factor 2C (MEF2C) nuclear translocation. Both events promote cardiomyocyte specification, transcription of sarcomeric proteins and myofibrillogenesis [97]. ROS also upregulate the expression of other cardiac-specific transcription factors and the expression of *Bmp10* [97]. During embryonic heart formation, BMP10 prevents premature cell cycle withdrawal of mesenchymal cells and regulates the level of expression of several key cardiogenic transcriptional factors [98]. After birth, BMP2 upregulates NOX4 and enhance ROS-driven p38MAPK activation which promote cardiomyocyte differentiation in stem cells [99]. All these events contribute to the modulation of cardiac growth and function.

BMPs also affect dendritic formation in sympathetic neurons [100], where they promote dendritic growth in hippocampal [101], cortical [102] and retinal ganglion neurons [103]. In sympathetic neurons, BMP7 upregulates *NOX2* expression, which is associated with an increase in oxygen consumption and ROS production. Newly synthesized ROS becomes restricted within certain cell compartments to avoid undesired cellular damage. In these sequestrations, they act as signaling molecules and upregulate the expression of differentiation genes [83]. Antioxidant treatment abolishes the effects of BMP7 on dendrite formation, data that demonstrates the importance of ROS in this phenomenon [81]. Additionally, BMP9 induces and maintains the neuronal cholinergic phenotypes in the central nervous system by inducing gene expression of choline acetyltransferase and vesicular acetylcholine transporter [104]. Interestingly, BMP9 levels and its bioavailability are reduced by redox-dependent proteolysis [105]. Consequently, increased extracellular ROS levels might be deleterious for neural differentiation. Although these data suggest a functional crosstalk between BMPs and ROS in the coordination of the differentiation of several cell types, further studies regarding the importance of the BMP and ROS interplay in the physiology of other tissues are necessary.

Cancer stem cells (CSC) constitute a small population of cells with the ability to proliferate indefinitely and give rise to the tumors [106]. In addition, differentiated progeny of CSCs create a niche that maintain both CSC and non-CSC components of the tumor hierarchy. BMPs have shown to be important not only for the differentiation of physiological stem cells but also for CSC. BMP2 have shown to induce CSC differentiation in osteosarcoma [107], while BMP4 induce differentiation of colorectal cancer stem cells [108], resulting in a reduction of tumorigenesis in both cases. In gliomas, where BMP2 expression is elevated, CSC secrete BMP antagonists, such as Gremlin1, to maintain their high proliferative capacity [109]. On the other hand, CSC, like normal stem cells have low intracellular levels of ROS, which contribute to their self-renewal capacity and resistance to chemotherapy. Several pathways are involved in controlling ROS levels in CSC. These pathways include PI3K/AKT/mTOR signaling pathway which regulate FOXO, HIF-1*α*, ATM, WNT, STAT and NF-*κ*B pathways [110].

## 8. BMPs and ROS in Pathology

### 8.1. ROS and BMPs in Vascular Diseases

Vascular vessels are structures integrated by endothelial, smooth muscle cells (SMCs) and fibroblasts that work in a coordinated manner to adapt to the changes in their environment. Vascular dynamics involve modifications in cell growth, death and migration, and the synthesis or degradation of the ECM. Vascular remodeling responds to physiological and pathological changes in haemodynamic conditions and is driven by the presence of local growth factors, vasoactive substances and hemodynamic stimulus [111].

ROS, as second messengers, induce vessel wall remodeling and influence SMC and endothelial cell growth and survival (Figure 4). Besides mitochondrial respiration, NOXs represent the major ROS source in vascular cells, and their dysregulation is associated with cardiovascular diseases [112]. The superoxide anion (O_2_^−^) is the major ROS molecule in the vascular wall. It inactivates nitric oxide (NO), a vascular relaxing factor, and thus impairs relaxation [113]. Several pathological conditions, such as inflammation or high blood pressure, activate NOX-derived H_2_O_2_ production via TNFα. In the endothelium, high pressure increases wall tension, which elevates the calcium concentration, activates PKC and stimulates NOX-dependent. O_2_^−^ and H_2_O_2_ production [114]. Increased H_2_O_2_ levels promote IκB degradation, and the consequent NF-κΒ activation induces *BMP2* expression [11,88]. Blood vessel smooth muscle and endothelial cells express *BMP2* and *BMP4* mRNA [88]; expression is higher in endothelial cells. Both *BMP2* and *BMP4* are upregulated in atherosclerotic lesions associated with oxidative stress, inflammation and hyperglycemia [115]. Moreover, plasma BMP2 levels positively correlate with calcium density of the plaque in hyperglycemia-induced vascular calcification [116].

In SMCs, BMP2 promotes the expression of the type III sodium-dependent phosphate cotransporter PIT1 via JNK pathway [117]. This effect is essential in matrix calcification induced by BMP-2. BMP2 also and increases the activity of NOX, which induces oxidative stress. The increase of NOX activity is mediated via activation of the bone morphogenetic protein receptor 2 and SMAD1 [118]. BMP2-mediated oxidative stress also induces endoplasmic reticulum (ER) stress that increases the function of GRP78, IRE1α, and XBP1. XBP1 is a transcription factor that binds to the Runx2 promoter and increases osteogenesis [118]. BMP2 also increases the levels of the osteogenic marker alkaline phosphatase (ALPL) and decreases the expression of the SMC marker *SM22*, changes that promote SMC transition to the osteochondrogenic phenotype and hence vascular calcification [117].

In endothelial cells, BMP2/4 activate PKC and MAPK pathways, which results in endothelial activation and increases monocyte adhesion. In these cells, MAPK activation is dependent on PKC and subsequent NOX-derived ROS generation [119]. In human coronary artery smooth muscle cells (HCSMC), BMP2 induce NOX via BMPR2 and SMAD1 [118]. In hypertensive patients, BMP2/4 upregulation is associated with impaired endothelial function in systemic blood vessels [120]. Moreover, BMP2/4 administration induces O_2_^−^ generation, endothelial dysfunction and hypertension in systemic arteries but does not affect pulmonary blood vessels [121]. In fact, in pulmonary arteries resistant to atherogenesis, BMP4 does not induce ROS production. Consequently, high circulating BMP4 levels selectively induce systemic (but not pulmonary) hypertension. In the endothelium of hypertensive patients, BMP4 upregulates *COX2* expression through a BMPR1A/p38 signaling pathway and stimulates NOX1 and NOX4 (but not NOX2), a phenomenon that increases ROS levels [122]. COX2 stimulation increases the release of constrictive prostaglandins, which promotes endothelium-dependent contraction. Induction of NOXs also increases the concentration of superoxide anions that inactivate NO and reduce endothelium dependent relaxations [123]. Besides, superoxide anions produced by NOX4 are converted to H_2_O_2_ by SOD. Then, H_2_O_2_ reacts with reduced iron ions, being converted into hydroxyl radicals that cause lipid peroxidation and protein carbonylation. Both modifications contribute to endothelial dysfunction [124]. Moreover, superoxide anions and H_2_O_2_ activate xanthine oxidase, which further promotes ROS production. All generated ROS initiate an inflammatory cascade and stimulates ICAM-1 expression leading to monocyte adhesion [82,125]. Thus, BMP4 inhibition has a potent anti-inflammatory and anti-atherogenic effect in coronary arteries [82].

In atherosclerosis mice models, BMP6 and oxidized low-density lipoprotein (oxLDL) synergistically regulate the expression of *Osterix* (*Sp7*) and *Osteopontin*. This process requires NOX activation and ROS production to induce osteogenesis and chondrogenesis in endothelial cells [126]. In the same way, oxLDL requires the activation of the BMP pathway for ROS production, and oxLDL induces the expression of other BMPs such as *BMP2* or *BMP9* [127]. Their combination with oxLDL or H_2_O_2_ potentiates canonical BMP signaling [126]. Similar effects are obtained in situations where BMP type II receptors are downregulated. Loss of BMPR2 decreases JNK signaling and enhances BMP9-induced mineralization. Both events contribute to the endothelial to mesenchymal transition (EMT) and vascular calcification [128]. Moreover, in models of cardiomyocyte hypertrophy generated by pressure overload or angiotensin II infusion, *Bmp4* expression increases and induces cardiomyocyte hypertrophy, apoptosis, and cardiac fibrosis in a NOX4- and ROS-dependent manner [129]. The pro-apoptotic activation of caspase-3 by BMP2 is triggered by NOX-derived ROS and downstream activation of p38 and JNK [120]. Consequently, BMP4 could be used as a predictor of vascular dysfunction and a pharmacological target to prevent vascular dysfunction [122].

In parallel, drugs that modulate oxidative responses are being investigated as an effective treatment for cardiovascular diseases. For instance, molecules that induce NRF2 transcription inhibit oxLDL production, hence decrease inflammatory cytokine generation and attenuate endothelial injury [130]. Activators of the PI3K/AKT signaling pathway are also useful in the treatment of cardiovascular disease, since they prevent ROS accumulation in the vessels and acts as a vasodilator [131].

### 8.2. ROS and BMPs in Obesity and Diabetes

Emerging evidence suggests that diabetes leads to depletion of the cellular antioxidant system, increases ROS levels and triggers oxidative stress that is involved in the complications associated with this disease [132]. The ROS sources in diabetes are still controversial. In vitro studies show that hyperglycemia enhances mitochondrial respiration and NOX activity. However, the main in vivo ROS source depends on the tissue and the type of diabetes. In type 1 diabetes mellitus (T1DM), mitochondrial ROS is implicated in cardiac complications, whereas NOX-derived ROS are associated with kidney and vascular injury [133,134]. However, in type 2 diabetes mellitus (T2DM), mitochondrial ROS are associated with kidney [135] and retina injury [136], whereas an increase in NOX activity is implicated in vascular tissue damage [137].

Obesity is associated with chronic inflammation and myocardial dysfunction. In the ventriculus of obese mice, *Nox1* and *Nox2* expression increases [138]. A similar pattern occurs for the expression of fibrotic markers, whereas anti-fibrotic markers BMP2 and active SMAD1/5 are significantly elevated upon reduction of obesity [138]. BMP pathway inhibition in obese mice reduces atherosclerosis and vascular calcification and improves lipid metabolism [127]. For instance, BMP inhibitors reduce apolipoprotein B100 secretion and thus lower the levels of LDL-cholesterol and improve liver steatosis [127]. In the liver, BMP inhibition reduces the levels of hepcidin, an iron regulator. Reduction of hepcidin by BMP inhibition reduces intracellular iron and oxidative stress [139]. In vivo mice models confirm that BMP pathway inhibition limits foam cell formation and atherosclerosis by reducing macrophage intracellular iron and lipid efflux capacity [139]. However, in hypertrophic obesity, upregulation of Gremlin, an antagonist of BMP4/7, impairs BMP4-induced beige/brown adipogenesis, inhibit the capacity of BMP4 to induce commitment and differentiation of preadipocytes and contribute to the dysregulation of the adipose tissue. Silencing *Grem1* and/or adding BMP4 during white adipogenic differentiation reactivated beige/brown markers [140].

In adipose tissue, NOX4-generated H_2_O_2_ modulates adipogenesis and adipose tissue function. Although ROS, as well as BMP4, mediate preadipocyte differentiation into adipocytes, in mature cells, elevated ROS levels induce adipocyte dysfunction. NOX4-derived ROS from adipocytes in the early stages of obesity initiate the recruitment of immune cells, including macrophages, and this phenomenon worsens insulin resistance and adipose tissue inflammation [141]. All these modifications converge in NOX upregulation and activation of NF-κB and p38 MAPK pathways that induce pro-inflammatory gene expression as well as excessive ROS formation in macrophages. Additionally, mitochondria-derived ROS partially maintain insulin resistance and adipose tissue inflammation during the late stages of obesity [141].

One of the main complications in diabetes is endothelial dysfunction. In T1DM, angiotensin II signaling activates NOX1 and induces tetrahydrobiopterin (BH4) deficiency and consequent endothelial nitric oxide synthase (eNOS) uncoupling [142,143]. However, in T2DM animal models, angiotensin II signaling is apparently not the main cause of endothelial dysfunction. Dyslipidemia contributes to elevate BMP4 levels in these animals. BMP4 can also activate NOX1 and induce BH4 deficiency. Further, BMP4 activates COX2 and vascular cell adhesion molecule 1 (VCAM-1) and induces inflammation in T2DM [143]. Diabetes is also associated with medial vascular calcification, also known as Mönckeberg’s calcification. This condition correlates with glucose levels in blood and the presence of calcium crystals in the medial layer [144]. Hyperglycemia and diabetes are strong activators of BMP signaling, and BMP2 and BMP4 are associated with atherogenesis in hyperglycemia. Moreover, in diabetic models, BMP4 disturbs blood flow and activates arterial NOXs, phenomena that promote ROS-induced inflammation, endothelial dysfunction and hypertension; BMP2 induces endothelial calcifications via WNT-MSX2 [145,146,147].

In the kidney, mesangial cells are the major source of glucose-dependent ROS generation. In the glomerular mesangial cells, oxidative stress caused by hyperglycemia activates the PKCζ isoform, which enhances TGFβ expression and increases ECM production. Overproduction of these components promotes morphological changes in the glomeruli. These changes hamper renal function and produce diabetic nephropathy, a serious complication in diabetes [148]. In recent years, recombinant BMP7 emerged as a potential therapy against diabetic nephropathy. BMP7 inhibits the phosphorylation and activation of PKCζ and JNK and reduces ROS generation and ECM synthesis. BMP7 also rescues podocytes from high-glucose-induced apoptosis by regulating SMAD5 and p38 phosphorylation [149]. Interestingly, high glucose levels induce the expression of *Gremlin*, a BMP antagonist. In mesangial cells, Gremlin induces transdifferentiation of tubular epithelial cells to a fibroblast-like phenotype [150]. In retinal pericytes and in the vascular walls, gremlin antagonizes the antiproliferative effects of BMPs. This action contributes to increased vascular endothelial proliferation and ultimately proliferative retinopathies [151]. Oxidative stress is another key event in the pathogenesis of diabetic retinopathy. In the retina of diabetic patients, the activity of antioxidant enzymes is diminished. The increased concentration of ROS, besides directly affecting cellular structures, activate other metabolic pathways that are detrimental to the development of diabetic retinopathy including PKC [152]. Consequently, modulation of BMP signals and ROS scavenging might serve as targets for high-glucose-associated pathologies.

### 8.3. ROS and BMPs in Kidney Injury

Sepsis, ischemia and nephrotoxicity are causative of acute kidney injury (AKI), which promotes structural damage and loss of kidney function. When treated quickly, the damage is reversible and kidney function is restored. However, in some cases AKI can lead to chronic kidney disease (CKD), where kidney damage worsens over time and kidney failure could occur [153]. Other conditions such as diabetes, hypertension or heart disease can also cause CKD. A characteristic of renal chronic diseases is the accumulation of excessive ECM and the presence of myofibroblast within the interstitium [154].

In the kidney, *BMP2* is expressed by metanephric mesenchymal cells, mesangial cells in the glomerulus and differentiated podocytes. Moreover, in vivo AKI models demonstrate *Bmp2* upregulation in adult renal progenitor cells, where it induces *Sma*, *Col1a1* and *Fibronectin* expression, all of which drive a myofibroblastic transition. Furthermore, in this cell type BMP2 promotes NOX4 expression and activity. NOX4 is also necessary for TGFβ-induced kidney myofibroblast activation and to promote the transition of fibroblasts to myofibroblasts [78]. In podocytes, BMP2 increases the Ca^2+^ concentration and stimulates PKC activity, which is involved in the activation of NOX family members. In podocytes, BMP2 also up-regulates ID1, an inductor of NOX activity [83]. Thus, BMP2 stimulates ROS production through NOX enzyme activation in adult renal progenitors and podocytes. Elevated ROS levels induced by BMP promote the EMT, contribute to increased ECM production and hence impair kidney function.

On the other hand, BMP7 negatively regulates TGFβ1 signaling and suppresses inflammation, apoptosis and EMT. Moreover, BMP7 has antifibrotic activity, which prevents ureteral obstruction and diabetic nephropathy [155]. In chronic kidney injury, BMP7 can reverse TGFβ1-induced EMT and repair severe damage to renal tubular epithelial cells, actions that reverse chronic kidney injury [156]. Therefore, BMP7 signaling agonists play a protective role against renal fibrosis in kidney injury models and are postulated as a potential therapy in renal diseases [157,158]. In a similar way, silencing BMP7 antagonists, such as Gremlin, can induce therapeutic effects. For instance, in biopsies from patients with diabetic nephropathy, the levels of Gremlin are upregulated and correlated with elevated serum creatinine levels and tubulointerstitial fibrosis [159]. Inhibition of Gremlin allows the efficient binding of BMP7 to its receptor and improves kidney injury [160]. Furthermore, elevation of the antioxidant system by NRF2 activators appears to be beneficial in renal fibrosis [161,162].

### 8.4. ROS and BMPs in Musculoskeletal Diseases

Sarcopenia is a syndrome characterized by progressive and generalized loss of skeletal muscle mass and function. Histologically, sarcopenic muscles are characterized by the presence of adipose infiltration, fibrotic tissue and a decline in type II muscle fiber satellite cell content [163]. The cell fate decision between adipocyte and myoblast depends on BMP7 levels. In Myf5+ progenitors, BMP7 represses the expression of the adipogenic inhibitors *Necdin* and preadipocyte factor 1 (*Pref1*) and induces the expression of *Prdm16*, which activates the brown adipogenesis program and blocks the induction of myoblast-specific genes such as *Myf5* and *Myod1* [164]. Consequently, when BMP7 levels are high, Myf5+ progenitors differentiate into brown adipocytes, whereas if *BMP7* expression decreases, Myf5+ progenitors can differentiate into the myogenic lineage [165].

In MSCs, chronic oxidative stress leads to S100 calcium-binding protein B (S100B) accumulation, increased NF-κB transcriptional activity and reduced expression of the promyogenic and anti-adipogenic mircoRNA-133 (*miR-133*). *miR-133* reduction upregulates *PRDM-16*, which stimulates brown adipogenesis. In parallel, NF-κB activity also upregulates BMP7 expression that stimulates the myoblast-brown adipocyte transition in an autocrine/paracrine manner [166]. Thus, ROS-mediated BMP7 upregulation might promote sarcopenia, and its inhibition can be beneficial for this disease.

Osteonecrosis arises from the death of bone cells due to decreased blood flow. When osteonecrosis occurs next to the joints, it can lead to osteoarthritis (OA). In bone cells, the hypoxia produced by osteonecrosis increases the production of free oxygen radicals, including O_2_^−^ and H_2_O_2._ Increased *BMP2* expression occurs in necrotic lesions [167] and damaged cartilage from OA patients linked to higher ROS production [168]. Moreover, serum BMP2 levels correlate with OA severity, hence it has been suggested that its levels could also be used as a biomarker for this disease [169,170]. Interestingly, after ischemic osteonecrosis, hypoxic chondrocytes within the cartilage transcriptionally activate the *BMP2* promoter. BMP2 stimulates proteoglycan synthesis, induces vascularization, promotes endochondral osteogenesis and has anabolic effects on chondrocyte metabolism and function [168,169]. In osteoarthritic cartilage, the increased concentration of BMPs is followed by an up-regulation of BMP antagonists, such as *Gremlin* and *Follistatin. Gremlin1* expression is elevated since the early development of OA and its levels correlate with the progression of the disease [171,172]. *Gremlin1* is also highly expressed in synovia and synovial fluids of patients with RA (Rheumatoid Arthritis) and its levels correlate with the concentration of proinflammatory cytokines. Upregulation of *Gremlin1* induce the activation of ERK1/2, AKT, and increased expression of *BCL2* while it reduces the levels of *BAX*. Both facts increase the survival, and capacity of migration of fibroblast-like synoviocytes [173] Treatment with Gremlin1 markedly inhibited terminal hypertrophic differentiation with no effects on the chondrogenesis [174] in conclusion, Gremlin1 could be a good therapeutical target in RA, capable of demoting hyperplastic synovitis [173].

## 9. Conclusions

Many biochemical, pharmacological and genetic studies confirm the relevance of ROS and BMPs in numerous aspects of cell biology and pathophysiology. Evidence also supports that both signals influence each other: BMP-dependent regulation of ROS production, and vice-versa, increases in the expression of distinct BMPs or modification on the expression of BMP antagonists in response to ROS generated by oxidative stress or NOX activity. However, many questions are still open and much more remains to be learnt regarding the relevance in the interplay between ROS and expression of specific BMPs and their antagonists in the context of normal tissue homeostasis and in pathological conditions. It should be possible to determine the effects of genetic alterations in the BMP signaling on cellular ROS levels or whether the cellular dysfunction associated to oxidative stress also depends on alterations of BMP expression. Additionally, BMPs and ROS share a number of downstream signaling components, suggesting a putative signaling crosstalk between the pathways. For instance, BMP and ROS activation of p38 MAPK or PI3K/AKT are well reported. Such parallel behavior and close signaling connections suggest that BMPs, BMP antagonists and ROS might regulate in coordination cellular physiology. However, despite this compelling rationale, strong evidence that such mechanisms are relevant is still lacking. Elucidation of these molecular mechanisms will not only advance basic biology, it may also provide new strategies for the treatment of a number of pathologies.

## Figures and Tables

**Figure 1 biomolecules-09-00534-f001:**
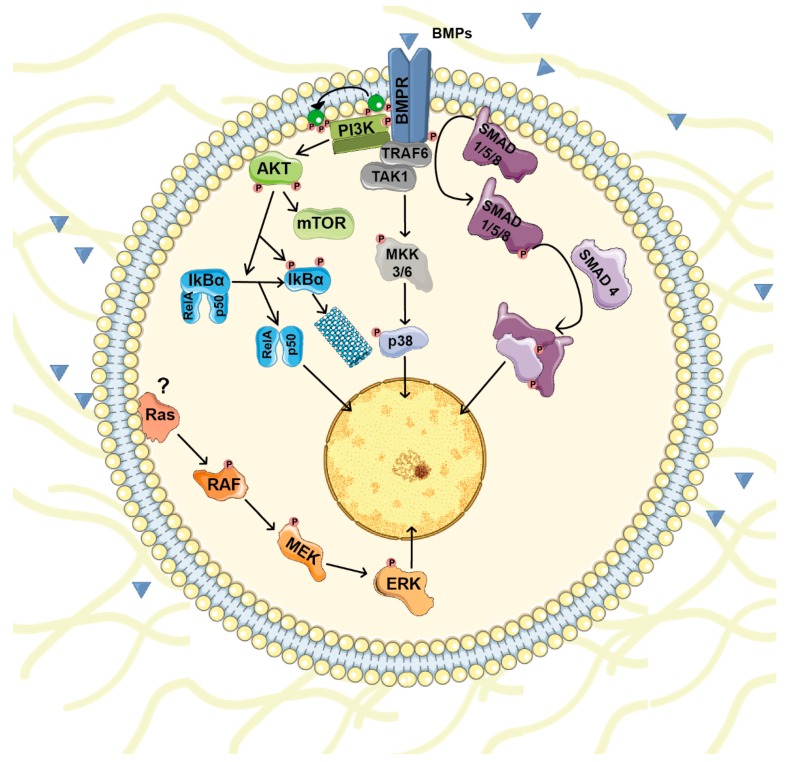
BMP signaling pathways. Once BMP receptors (BMPR) are stimulated they form a complex and initiate an intracellular transduction cascade. In the canonical pathway, BMP receptors phosphorylate and activate the SMAD family of transcription factors that transduce the signal to the nucleus. Besides SMADs, BMPs modulate other intracellular signaling pathways (non-canonical BMP signaling) including PI3K/AKT and various types of MAPKs pathways.

**Figure 2 biomolecules-09-00534-f002:**
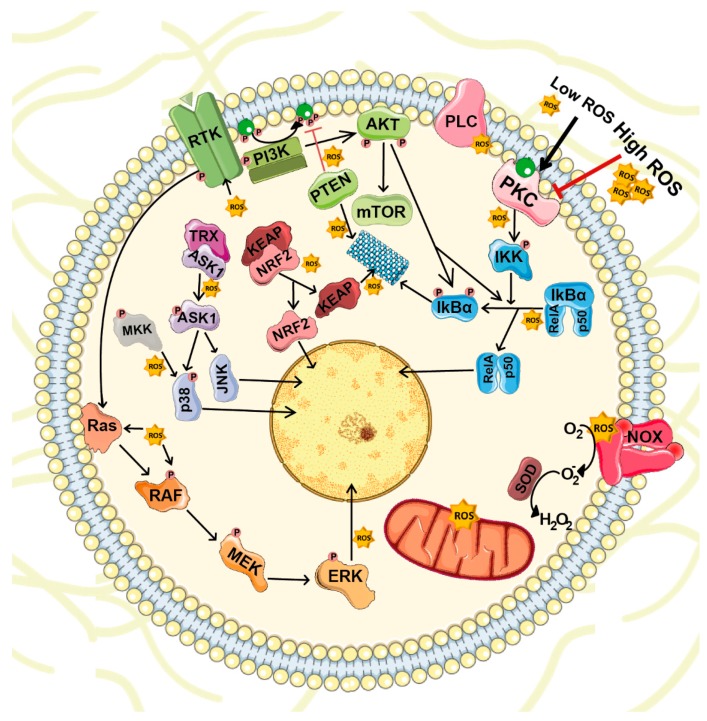
ROS-mediated cellular signaling. ROS are highly reactive molecules that act as second messengers inside the cell. Besides mitochondrial respiration, NOXs represent the major ROS source. Under physiological conditions, intracellular ROS target and stimulate several transductor proteins NF-κB, MAPKs, KEAP1-NRF2 and PI3K-AKT important for cell survival, proliferation and differentiation.

**Figure 3 biomolecules-09-00534-f003:**
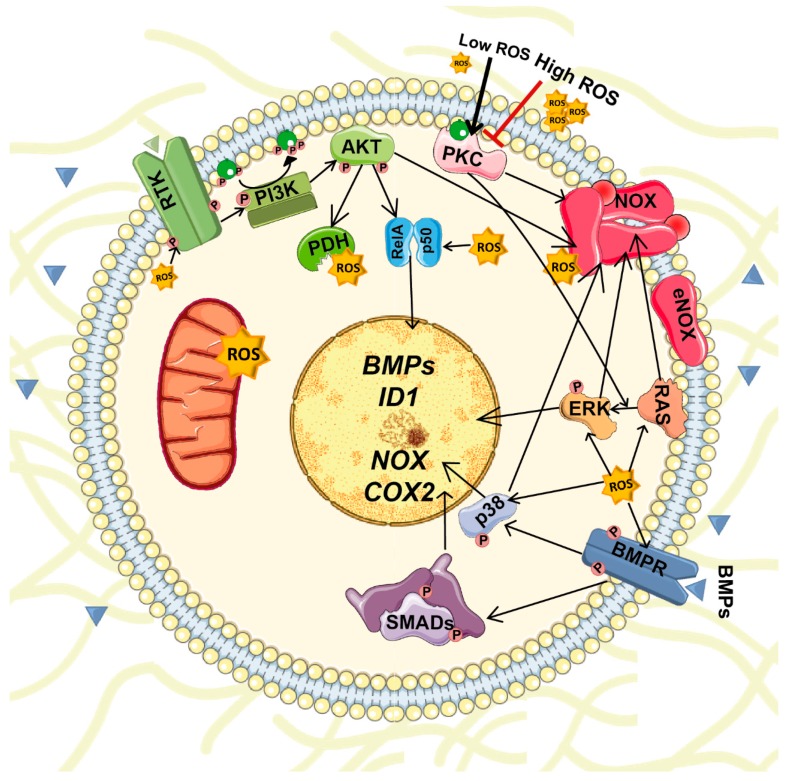
Interplay between BMPs and ROS signaling. Production of BMPs and ROSs are reciprocally regulated. BMPs stimulate ROS production by activation and/or induction of NOX1-5 expression while ROS regulate the expression of several BMPs. Furthermore, both BMPs and ROS activate several common signaling pathways including PI3K-AKT, ERK, JNK and p38 MAPKs.

**Figure 4 biomolecules-09-00534-f004:**
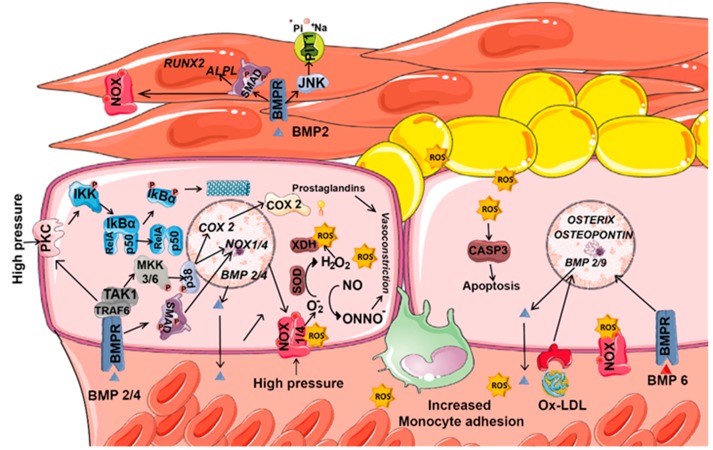
Coordinated contribution of ROS and BMPs to vascular diseases. Vascular vessels are structures integrated by endothelial, smooth muscle cells (SMCs) and fibroblasts. In the endothelium, ROS are mainly produced by NOXs and its levels increase as a response to high pressure. In vascular vessels, ROS induce the expression of several BMPs, inactivates nitric oxide, and stimulate NF-κB pathway. By its part, BMPs activate MAPK pathways, increase *NOX1/4* and *COX2* expression and consequently enhance ROS generation. BMP6 and oxidized low-density lipoprotein (oxLDL) increase the expression of the osteoblast markers *Osterix* and *Osteopontin* in a NOX dependent manner. In SMCs, BMPs promotes the expression of the type III sodium-dependent phosphate cotransporter *PIT1*, increases the activity of NOX, and the levels of osteogenic markers (*RUNX2* and *ALPL*).

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
