# Peer review of "Interplay between BMPs and Reactive Oxygen Species in Cell Signaling and Pathology"

_biomolecules, 2019, doi:10.3390/biom9100534_

Round 1

Reviewer 1 Report

The paper reviews the current literature regarding the interplay between BMP and ROS signaling. The paper does a good job of providing the known signaling pathways that regulate BMP and ROS or are regulated by them. However, in certain areas of the review, the interplay between these pathways is not clear based on the data provided. Also, it would be good to articulate whether these pathways are synergistic or antagonistic in the various contexts. So, I would recommend a revision/reorganization of the text.

Here are some specific concerns/comments.

The paper starts with an introduction of BMPs. In later sections, the authors refer to TGF signaling and its interplay with ROS. However, there is no reference in the introduction or earlier sections about BMPs being a member of the TGF-beta superfamily. This would be a useful addition. The introduction of BMP is limited to the role of BMP in embryogenesis. However, the examples used later for interplay between ROS and BMP do not talk about embryonic patterning. It would be good to include the role of BMP in nervous system and vascular development in this section. Paragraph starting on line 56 – needs a reference for the first couple of sentences which talk about ROS and NOX. Line 97 – refers to GS domain, however, there is no information about the importance of GS domain or what GS stands for. Similar On Page 9 – section 6, This section refers to the interplay between BMP and NOX. The section does not indicate the types of cells where each of these effects is observed. On line 297 – the authors state that BMPs induce COX2, which is a source of ROS. Need a reference for the statement for COX2 is a source for ROS. The sentence starting on line 299 could be taken as BMPs activate PI3K and this activation has been to shown to elevate ROS or that BMPs activate PI3K and in a different study PI3K has been shown to regulate ROS. So, it needs further clarification and reference to determine which interpretation is correct. Titles of section 8 – 8.1 includes ROS and BMP in title, others don’t. In figure 4 – BMPs are shown to affect NOX but there is no arrow from BMPs to COX2, although the text refers to upregulation of COX expression by BMP. It is not clear from the text and the image if the current data indicate that BMP effects on NOX and COX2 are parallel pathways or one leads to the other. Clarification – line 450 – does the loss of BMPR2 lead to changes in NOX or ROS? If the data are not there, it would be good to state something like – ‘Although there is no direct correlation between loss of BMPR2 and ROS, studies have shown that BMPR2 loss can decrease JNK signaling … a pathway shown to regulate ROS. Section 8.2 – Lines 477-487 : The interactions between BMP and ROS are not clear. The sentence on 484-485 states that “reduction in hepcidin…. reduces oxidative stress” – there is no reference for this statement. Is this shown in the ref 125? Lines 498 – 511 –Statements on line 503 and line 508 about BMPs and BH4 deficiency and atherogeneis, respectively – need references. The interplay between BMP and ROS in mesangial gells (lines 522 – 527) is not clear. The role of BMP is stated in retinopathy. What about ROS? Section 8.3 – line 541 – refers to TGF-beta turning on NOX4 but the relationship between TGF and BMP is not mentioned earlier. Under conclusions, I would suggest expanding the lines 598 – 600 to be a section that addresses the gaps in the current data and the areas of research that would help further understand the interaction between these two pathways.

Author Response

Reviewer #1.

We introduced the suggestions of the reviewer throughout the manuscript. We emphasized the fact that BMPs belong to the TGF-β superfamily and expanded the introduction of BMPs beyond embryogenesis adding information about the role of BMPs in vascular and nervous systems. We also modified throughout the manuscript all the paragraphs suggested by the reviewer by including: clarification or addition of appropriate citations in some sentences. Moreover, we modified Figure 4 to show the effects of BMP signalling in NOX1 and COX2 expression and expanded the conclusions.

We acknowledge the reviewer comments and suggestions that strengthened the main conclusions of our manuscript. We think that this revised version addresses all the concerns of the referee. We thank you for your careful revision of our manuscript.

Reviewer 2 Report

This is very comprehensive review about the interplay between BMPs and ROS in cell signalling. This is an important topic as there is a wealth of evidence that BMPs and ROS play significant roles in human disease, and likely interact at multiple points in the signalling cascades.

I have the following minor comments:

Page 2 lin 36 replace "suggest" with "demonstrate" Page 3 line 86-88-provide more detail on the antagonists, with other references (e.g. PMID 25592806, 20545624) Page 3 line 89 replace "in the cell surface" with "outside the cell" Page 7 line 191-192 include detail on pThr308 phosphorylation of Akt by mTORC1 and DNA-PK Page 12-mention the role of BMP-2 and Grem1 in cancer stem cells (CSCs), and the role of BMP-2 in driving CSC differentiation Page 15 line 470-472 mention the papers from the Ulf Smith lab (e.g. PMID 25605802) Page 17 line 522 replace "inhibitor" with "antagonist" Page 17 section 8.3 references on the role of BMP antagonists such as Grem1, USAG1, Noggin etc in kidney injury such as diabetic nephropathy Page 18 line 576-580 mention the paper on the role of Grem1 in synovial arthritis (e.g. PMID 26834210) In the Conclusions, perhaps allude to the importance of the balance between BMP and antagonists in physiology, and how imbalances in these proteins are implicated in human disease. 

Author Response

Reviewer #2.

We introduced the suggestions of the reviewer throughout the manuscript. We modified the manuscript as suggested by the reviewer by including: clarification of some sentences or addition of appropriate citations. We also expanded the information about the role of BMPs in cancer stem cell biology and AKT activation. Moreover, we expanded the information about the role of BMP antagonists in the manuscript and conclusions.

We acknowledge the reviewer comments and suggestions that strengthened the main conclusions of our manuscript. We think that this revised version addresses all the concerns of the referee. We thank you for your careful revision of our manuscript.

Round 2

Reviewer 1 Report

The authors have satisfactorily addressed my comments.